# Unsupervised Deep Learning Method for MRI Bias Correction

**Maria Perez-Caballero**[*1]                                    MPERCAB1@ETSINF.UPV.ES
[1] *Instituto de Aplicaciones de las Tecnologías de la Información y de las Comunicaciones Avanzadas (ITACA), Universitat Politècnica de València, Camino de Vera s/n, 46022, Valencia, Spain*

**Sergio Morell-Ortega**[1]                                    SERMOOR1@TELECO.UPV.ES
**Marina Ruiz-Perez**[1]                                        MRUIPER@ETSII.UPV.ES
**Pierrick Coupé**[2]                                           PIERRICK.COUPE@GMAIL.COM
[2] *CNRS, Univ. Bordeaux, Bordeaux INP, LABRI, UMR5800, in2brain, F-33400 Talence, France*

**José V.Manjón**[*1]                                           JMANJON@FIS.UPV.ES

**Editors:** Accepted for publication at MIDL 2024

## Abstract

In this paper, a new method for automatic MR image inhomogeneity correction is proposed. This method, based on deep learning, uses unsupervised learning to estimate the bias corrected images minimizing a cost function based on the entropy of the corrupted image, the derivative of the estimated bias field and corrected image statistics. The proposed method has been compared with the state-of-the-art method N4 providing improved results.

**Keywords:** Unsupervised, Deep Learning, MRI

## 1. Introduction

In magnetic resonance imaging, the presence of signal intensity inhomogeneity, often referred to as the bias field artifact, poses a notable challenge. Originating from imperfections in radiofrequency coils and specific interactions within the imaged object, this artifact manifests as a low-frequency variation in signal intensity across MR images. While it may not significantly impact qualitative diagnoses, its influence on automated quantitative methods, particularly registration and segmentation, is of considerable importance.

Addressing this artifact involves both prospective and retrospective strategies. Prospective methods aim to prevent inhomogeneity during the acquisition process, primarily focusing on mitigating machine imperfections. In contrast, retrospective methods, which have undergone more extensive development, do not require special acquisition protocols and can effectively handle both machine and patient-induced inhomogeneities. These retrospective methods can be further categorized into segmentation-based approaches, where the bias field is estimated during segmentation, and direct data methods that work directly with the image data.

Various approaches have been proposed to tackle intensity inhomogeneity, among which SPM and histogram-based methods are noteworthy. SPM, (Ashburner and Friston, 2000), addresses intensity inhomogeneity by modeling the bias field using a combination of Discrete Cosine Transform basis functions. The parameters of these basis functions are adjusted

---

* Contributed equally

through the minimization of the negative log-likelihood of the data, which is equivalent to optimizing image entropy. However, limitations highlighted b (Arnold et al., 2001) pointed out that entropy minimization tends to favor a uniformly zero bias field, resulting in a single bin image. This issue was subsequently addressed in SPM2 (Ashburner, 2002) by utilizing log-transformed image intensities for entropy calculation. On the other hand, histogram-based methods, such as the pioneering work by Brechbuhler et al. (1996) and the widely recognized N3 method (Sled et al., 1998), leverage statistical properties of image histograms to correct intensity biases. Brechbuhler et al. (1996) estimated the bias field using Legendre polynomial basis functions and the minimization of a cost function based on the means and variances of the tissues in the corrected image, while the N3 method sharpens the image histogram through Gaussian deconvolution and smoothens the bias field estimation using a B-Spline based regularization. These approaches showcase the diverse strategies employed to mitigate intensity inhomogeneity, each with its strengths and considerations. Significant strides have been made in the realm of bias field correction for medical images, with notable advancements introduced by Mangin (2000), Likar et al. (2001) and Manjón et al. (2007). Manjón et al. proposed an innovative method employing a coarse-to-fine strategy and a joint intensity-gradient entropy cost function, effectively addressing issues of regularization. Likar et al. (2001) and Mangin (2000) proposed solutions involving constraints on restored image mean values, providing valuable alternatives in mitigating entropy minimization-related challenges. Despite these advancements, the N4 (Nonparametric Non-uniform Intensity Normalization) method has emerged as a predominant and widely adopted approach in current practices. Serving as an extension of the N3 algorithm, N4, introduced by Tustison (Tustison et al., 2010), employs nonparametric techniques for robust bias field correction in medical images, particularly in MRI. Its iterative approach in estimating and correcting non-uniform intensity has proven highly effective, offering robustness in addressing bias fields of varying amplitudes. The method's adaptability and reliability have positioned it as a go-to choice in contemporary medical image analysis, underscoring its impact and practicality in addressing the multifaceted challenges associated with bias field correction in the field of medical imaging. More recently, some deep learning approaches have been proposed for the inhomogeneity correction problem. Among them, ABCnet (Chen et al., 2021) proposes a 3D adversarial bias correction network for infant brain MR images, incorporating manual corrections and leveraging GANs for enhanced efficiency and intensity uniformity. Another approach (Xu et al., 2022), utilizes deep separable convolutional neural networks with local feature images, integrating residual learning and batch normalization. This paper presents an innovative unsupervised bias correction method based on deep learning: The Unsupervised Bias field Corrector (UBC). The method, explained in the following sections, employs unsupervised training based on an entropy and gradient related loss, demonstrating excellent both time efficiency and accuracy results.

## 2. Material and methods

The acquired MR image model to describe MRI bias corrupted data is a multiplicative field plus some additive noise:

$$Y = XB + n. \tag{1}$$

where Y is the observed voxel intensity, B the corresponding value of the bias field supposed to be smooth, X the true emitted intensity and n is a Rician distributed additive noise. Inhomogeneity correction thus consists of dividing Y by an estimation of B by using the common assumption that this bias field is positive and slow varying. The method proposed in this paper has been developed using 2 public MR datasets:

- OASIS (N=374): These T1 MR images are sourced from the Open Access Series of Imaging Studies (OASIS) database (http://www.oasis-brains.org).

- IXI (N=580): The images from the Information eXtraction from Images (IXI) database (http://brain-development.org/ixi-dataset) consist of normal subjects T1 images scanned at 1.5T and 3T.

The training set was composed of 914 T1 MR images (354 from OASIS and 560 from IXI) and the test set was composed of 40 images (20 IXI images and 20 OASIS). The images were normalized by dividing by the mean value.

## 3. Neural Network Architecture

The architecture of a deep learning model plays a pivotal role in its ability to comprehend and process complex patterns within data. The proposed architecture consists of an encoder part, focused on capturing image features and a decoder part centered on the estimation of the bias field. In the encoder, four convolutional layers (with ReLU activation and 3x3x3 kernel size) are used, each one with its pooled output (stride=2). The inclusion of batch normalization enhances stability. This convolutional sequence commences with 16 filters, progressively doubling, quadrupling, and octupling in subsequent layers—16, 32, 64, and reaching a crescendo with 128 filters. The progressive increase in filter counts in successive convolutional blocks enhances the model's capacity to discern nuanced features within the 3D images. Strided convolutions with a factor of 2 in the encoder layers facilitate downsampling, reducing spatial dimensions while preserving essential features. At the end of the encoder a positive definite tensor of 1x7x7x7x1 elements is obtained representing the control points of the estimated bias field. To estimate the bias field, this tensor is interpolated through an upsampling layer to match the size of the input tensor. Finally, to obtain the corrected input tensor the estimated bias field is multiplied by it (note that for stability we multiply by the inverse of the bias field instead of dividing it to avoid zero division instability). To evaluate the quality of the corrected image a histogram estimation layer is used. This custom layer uses a kernel density estimation (KDE) approach to estimate the image entropy loss. Other auxiliary losses are used to regularize the estimated bias field (Figure 1 shows a visual representation of the proposed architecture).

In summary, our architecture stands as a solution for 3D medical image bias correction, integrating advanced convolutional techniques, normalization strategies, and specialized layers to ensure accurate bias field estimation and high-quality image correction.

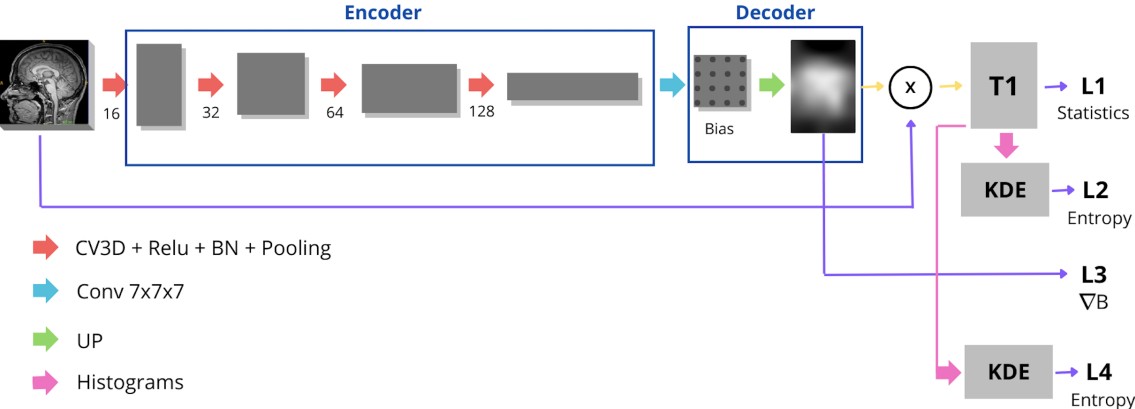

Figure 1: Proposed UBC bias correction architecture applied to a T1 input

## 4. Training

As this method works at native MR image space, the first step is to resample the input image to a fixed size of 100x100x100 voxels. The estimated bias field is interpolated back to the size of the native space with a minimum loss of accuracy due to the inherent low frequency content of the bias field. Input images were normalized by dividing them by their mean. KDE layer uses a gaussian function with a standard deviation of 1 and 200 bins.

As seen in the introduction section, the reduction of image entropy has been largely used as a metric to estimate bias fields. The main assumption is that homogeneous brain tissues generate sharper peaks in the histogram which is related to the entropy reduction through a more homogeneous intensity order. However, such an approach solely can result in a single bin histogram which is the minimum entropy solution. To avoid this issue, some regularization is needed. In our training, we used a mix of 4 specific loss functions, each tailored for a particular role in guiding the bias correction task.

The first is the entropy loss which is directly estimated from the histogram layer output. To restrict the solutions space, a global statistics loss is used which penalizes that the standard deviation of the corrected image differs too much from the original one and forces the output image to have a mean value of one (same as the input). To enforce the low frequency content of the estimated bias field the expectation of local gradients was used as loss function. This loss penalizes high local derivatives of the estimated bias field and plays a crucial role in promoting smooth transitions within the bias fields, contributing to a visually coherent correction. By prioritizing gradient smoothness, the model ensures a more natural and faithful representation of the underlying anatomy. Finally, the last loss is the entropy of the local gradients of the corrected image. In Manjón et al. (2007), the minimization of the joint intensity-gradient entropy was used but in architecture we had to separate the intensity and gradient entropy losses due to memory constraints.

Balancing the loss functions (Equation (2)) was achieved through experimental assignment of weights, acting as a crucial mechanism during training. This ensures each loss

contributes optimally without overpowering others, preventing biases that could distort the inherent characteristics of the images.

$$\text{loss} = \lambda_1 \cdot H(Y) + \lambda_2 \cdot H(\nabla(Y)) + \lambda_3 \cdot \nabla\overline{B} + \lambda_4 \cdot (|\sigma_p - \sigma_y|) \tag{2}$$

During training data augmentation was used consisting in random flipping the volumes along the 3 dimensions.

## 5. Experiments and results

In this section, we present a summary of the experimental outcomes aimed at identifying the optimal network. The experiments were conducted using Tensorflow 2.10 on a RTX 3060 GPU with 12GB operating on Windows 10, each network tested underwent training for 500 epochs, equivalent to a day of processing, ensuring convergence. The evaluation of the resulting networks was carried out on a test dataset comprising 40 cases. Adam optimizer was used in the experiments using mixed precision settings. Several architectures were tested, mainly varying the number of control points of the bias field (3x3x3,5x5x5,7x7x7,9x9x9, and 11x11x11 control points were tested) as this is directly related to the smoothness of the obtained bias field. As a result, we selected the 7x7x7 architecture as it asd the one that obtained the best results. Regarding the loss balancing, the best results were obtained for $\lambda_1 = 2$, $\lambda_2 = 1$, $\lambda_3 = 1$ and $\lambda_4 = 1$.

To assess our model's effectiveness, we employed the coefficient of joint variation (CJV) metric (Manjón et al., 2007). To calculate the coefficient of joint variation for both gray and white matter, segmentation maps were obtained using vol2Brain method (Manjón et al., 2022). The coefficient of variation (CV) of GM and WM tissues was also used. The objective was to minimize the pixel-wise variation within each type of tissue. This evaluation method provides a comprehensive measure of our model's performance, emphasizing the reduction of variation between pixels within both gray and white matter and tissue mean contrast.

In an attempt to enhance results, two additional steps were used at test time. The first one is known as Test Time Data Augmentation (TTDA) and consists of evaluating the bias field after applying a flip operation, one for each axis, to the input image and later inverting them and averaging the predictions. The second, given the fact that the method is unsupervised, was to perform additional training steps using only the input image to be corrected (fine tuning). We call this step TTT (Test Time Training).

The results of the proposed method for the different configurations is summarized at Table 1. As can be noted, both TTDA and TTT improved the results of the origin method at the expense of increasing the temporal cost.

Table 1: Test results of the proposed method.

| Method | CJV | CV(WM) | CV(GM) | Time (s) |
|---|---|---|---|---|
| Original | $0.8565 \pm 0.2135$ | $0.1195 \pm 0.0229$ | $0.2020 \pm 0.0262$ | - |
| UBC | $0.7697 \pm 0.0819$ | $0.0920 \pm 0.0205$ | $0.1789 \pm 0.0312$ | $0.26 \pm 0.25$ |
| UBC+TTDA | $0.7686 \pm 0.0816$ | $0.0920 \pm 0.0204$ | $0.1787 \pm 0.0311$ | $0.40 \pm 0.26$ |
| UBC+TTDA+TTT | $0.7616 \pm 0.0808$ | $0.0923 \pm 0.0221$ | $0.1790 \pm 0.0322$ | $17.02 \pm 0.48$ |

We compared the proposed method with the state-of-the-art N4 method as implemented in the SimpleITK library. Results are summarized in Table 2. The proposed method improved the N4 method while being 5 times faster. Although the differences of the proposed method and N4 are not statistically significant it seems to be more robust than N4 producing lower minimum and maximum CJV values. Besides, if we don't use the TTT step, the proposed method is 222 times faster with similar results than N4.

Table 2: Comparison with N4 method. Mean and standard deviation of CJV metric is used.

| Method | CJV | CV(WM) | CV(GM) | Time (s) |
|---|---|---|---|---|
| Original | $0.8565 \pm 0.2135$ | $0.1195 \pm 0.0229$ | $0.2020 \pm 0.0262$ | - |
| UBC+TTDA+TTT | $0.7616 \pm 0.0808$ | $0.0923 \pm 0.0221$ | $0.1790 \pm 0.0322$ | $17.02 \pm 0.51$ |
| N4 | $0.7646 \pm 0.0838$ | $0.0947 \pm 0.0253$ | $0.1800 \pm 0.0345$ | $89.25 \pm 27.66$ |

A visual example of the output of the proposed and N4 methods is illustrated in Figure 2. As can be observed, both N4 and UBC produce sharper histograms but UBC produces taller peaks representing a lower dispersion of image intensities.

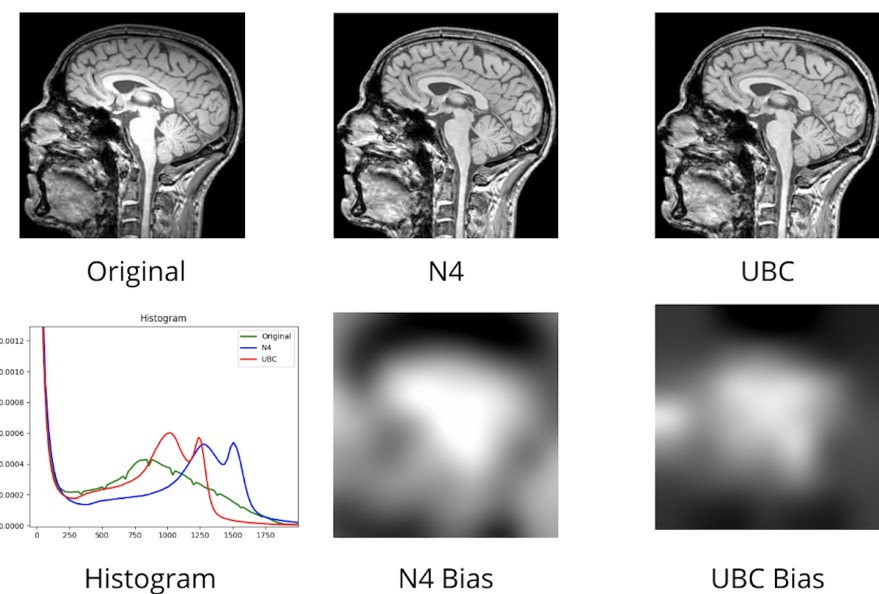

Figure 2: Comparison between Original MRI, N4 and UBC corrected images. The estimated bias fields and the histogram of each image are also shown.

To demonstrate the generality of the proposed unsupervised method, we also trained our method using T2 weighted images. Specifically, we used 556 images for training and 20 for testing from the IXI dataset. Results can be checked in Table 3. As can be noted, the proposed method slightly improves the N4 method while being much faster.

Table 3: Comparison between UBC method and N4 method for T2 images.

| Method | CJV | CV(WM) | CV(GM) | Time (s) |
|---|---|---|---|---|
| Original | $1.5308 \pm 0.1438$ | $0.2643 \pm 0.0241$ | $0.3128 \pm 0.0335$ | - |
| UBC+TTDA | $1.4874 \pm 0.1414$ | $0.2413 \pm 0.0294$ | $0.3082 \pm 0.0397$ | $0.42 \pm 0.34$ |
| N4 | $1.4917 \pm 0.1412$ | $0.2453 \pm 0.0316$ | $0.3074 \pm 0.0403$ | $20.04 \pm 11.00$ |

## 6. Discussion

In this paper, we present an unsupervised deep learning based MRI inhomogeneity correction method. The outcomes delineated in the preceding section underscore the significant advancements achieved through the presented methodology. Not only do these results highlight a substantial decrease in brightness variability among pixels belonging to the same tissue, but they also emphasize the capacity to attain this improvement without compromising the authenticity of the original image features. This preservation ensures that the image remains true to its initial characteristics, avoiding any form of distortion. The demonstrated success in minimizing bias and preserving image fidelity positions the proposed approach as a robust and effective solution in the domain of bias correction being competitive with conventional methods.

The proposed method, called UBC, is based on a simple, yet effective, architecture that encodes image features and produces an estimate of the inverse bias field minimizing a loss function based on features of the output image and the estimated bias field. In fact, although entropy minimization has been used in the past for MRI bias correction, this is, as far as we know, the first time that this metric is used in the context of a deep network for this task.

Incorporating test-time steps, TTDA and TTT, improved the results, enhancing our bias correction methodology. These additions have demonstrated their effectiveness in refining correction outcomes, contributing to consistently excellent results. Our approach slightly outperforms the conventional N4 bias correction method (although the difference is not statistically significant) and it is far more efficient.

We have trained our method using only T1 and T2 MR images to demonstrate its effectiveness. We are aware that the N4 method is by default a more general method than the proposed method. In the future, we will train UBC with all sorts of MR images (FLAIR, etc.) to get a more general solution.

## 7. Conclussion

The approach presented in this paper constitutes an innovative non-supervised method for bias correction, an area that has seen limited exploration. This novel non-supervised technique for MRI bias correction introduces a fresh perspective, showcasing its capability to outperform commonly used methods. This advancement is particularly significant as it demonstrates the feasibility of constructing a network that achieves highly effective results solely based on training input images, without relying on previous methods. The novelty

lies in the ability to devise a model that excels in bias correction without the need for additional information or dependence on traditional methodologies.

This novel non-supervised bias correction method not only demonstrates remarkable efficacy, being competitive with conventional approaches, but also distinguishes itself with exceptional efficiency. Notably, the model's capacity to yield superior results is complemented by its rapid processing time, enhancing its accessibility. This combination of effectiveness and efficiency positions the method as a promising advancement in MRI bias correction, showcasing its potential to significantly impact the field.

The Unsupervised Deep Bias Correction (UBC) method emerges as a swift and highly effective approach, capable of yielding outstanding results in a short timeframe. Its rapid yet precise correction of bias in MRI images positions it as a valuable tool for various applications, particularly in tissue segmentation processes. The robust performance demonstrated by UBC opens up avenues for future research in the realm of MRI image analysis. Its effectiveness and precision position it as a promising tool to improve the quality of data in medical imaging studies. This, in turn, facilitates more accurate tissue segmentation and contributes to advancements in MRI research.

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
