# OpenReview forum: "Unsupervised Deep Learning Method for Bias Correction"
_MIDL.io/2024/Conference — MIDL 2024 Poster_

### Official Review · Reviewer_DPdQ · 2024-02-26

**Confidence:** 4
**Preliminary Rating:** 2
**Recommendation:** Poster
**Final Rating:** 4

**Summary:**

The authors proposed an unsupervised deep learning method for MR bias field correction, with test time data augmentation and fine tuning. To evaluate the performance, metrics coefficient of joint variation (CJV), coefficient of variation (CV) of gray matter and white matter are used. Compared with N4 method, the proposed method showed better CJV and fast running time.

**Strengths:**

The paper is well-written and easy to understand. The author proposed an unsupervised deep learning method for MRI bias field correction, with four different losses to ensure the unsupervised training, which is novel. In addition, the authors applied test time data augmentation and fine tuning to increase performance during test time, which is a good point to address the test time performance drop when domain shift happened.

**Weaknesses:**

1. Multiple losses are used in the network training. The authors claimed that regarding the loss balancing, the best results were obtained for λ1 = 2, λ2 = 1, λ3 = 1 and λ4 = 1. However, it is unclear for readers the importances of each loss in the bias field correction. I would like to suggest to perform ablation studies to evaluate the effectiveness of each loss.
2. The authors failed to compare the proposed method with other deep learning methods, such as "MRI bias field correction with an implicitly trained CNN".

**Detailed Comments:**

1. In Table 2, CV(WM) and CV(GM) should be included for better comparision.
2. Table 1 showed that TTDA and TTT can improve the quantitative metrics, but slightly. From the paper, the training and testing dataset are from the same two open datasets. That could explain TTDA and TTT only slightly improve the metric scores. I suggest that the author to train the network on either OASIS or IXI, then fine tuning when testing on the other dataset, which could better show the effectiveness of TTT.
3. Perform ablation studies to evaluate the effectiveness of each loss.
4. The author can use downstream task such segmentation to compare the performance of different bias field correction methods.
5. The authors selected the 7x7x7 control points in the bias field base on experiment. I suggested to provide more details.

**Justification Of Final Rating:**

I appreciate authors' responses. I think the unsupervised learning for bias field correction is a good contributor to the conference. I found this paper, Simkó, Attila, et al. "MRI bias field correction with an implicitly trained CNN." International Conference on Medical Imaging with Deep Learning. PMLR, 2022. I think it is better to compare with other deep learning methods in the final submission.

**Justification Of The Preliminary Rating:**

Based on my best, I acknowledged that the authors proposed a novel unsupersed deep learning methods to perform bias field correction in MRI. However, the paper lacks necessary experiments and ablation studies, which needs to be addressed before acceptance.

**Questions To Address In The Rebuttal:**

1. Since they could be acquired using different scanner, sequence, imaging parameters, etc, some data pre-processing is needed before network training. Did you perform any data preprocessing on the OASIS and IXI T1 images? and How?
2. Any data augmentation during training phase?
3. How did you choose λ1 = 2, λ2 = 1, λ3 = 1 and λ4 = 1? Does it based on hyper-parameter tuning?

**Special Issue:**

No

---

> ### Author Response · Authors · 2024-03-12
>
> 1- In Table 2, CV(WM) and CV(GM) should be included for better comparision.
>
> We have added the information requested by the reviewer at tables 2 and 3.
>
> 2- Since they could be acquired using different scanner, sequence, imaging parameters, etc, some data pre-processing is needed before network training. Did you perform any data preprocessing on the OASIS and IXI T1 images? and How?
>
> No. We didn’t preprocess the images. We used the raw images into the native space. We acknolwledge that a prior denoising could further improve the results but this was not tested.
>
> “ The images were normalized by dividing by the mean value.”
>
> 3- Any data augmentation during training phase?
>
> Yes. In the text: “During training data augmentation was used consisting in random flipping the volumes along the 3 dimensions. “
>
> 4- How did you choose λ1 = 2, λ2 = 1, λ3 = 1 and λ4 = 1? Does it based on hyper-parameter tuning?
>
> No. We did it experimentally by trial and error.
>
> “Balancing the loss functions (Eq. 2) was achieved through experimental assignment of weights, acting as a crucial mechanism during training”

---

### Official Review · Reviewer_78wk · 2024-02-27

**Confidence:** 4
**Preliminary Rating:** 5
**Recommendation:** Oral
**Final Rating:** 5

**Summary:**

The manuscript „Unsupervised Deep Learning Method for MRI Bias Correction” proposes a unsupervised deep-learning based method to perform bias field correction in MR imaging. The presented work focuses on the class of retrospective bias field correction methods which act on acquired data and do not interfere with the data acquisition itself. After a good and sound introduction the authors describe their approach before presenting results in tabular form and with some example images. The discussion and conclusion summarize the results, but also have only little critical review of the presented work.

**Strengths:**

The authors address relevant challenge in MRI using timely methods and modern approaches. The results are promising and the introduced methods seem to allow relatively easy implementation of this unsupervised approach. Overall the authors have written a clear manuscript and present a piece of research which is valuable for discussion.

**Weaknesses:**

An old fashion standard to quantify bias fields in multi-coil receivers i.e. standard head coils is to use the body coil for reference. This could be a piece of future research and should be considered also to distinguish between different contributions to the bias field. Also, it would be very interesting to see the performance of the presented approach in severely biased images, e.g. images with metal implants visible. The authors should consider such cases in their future work which would increase scientific interested and impact a lot.

**Detailed Comments:**

-	Please indicate that Eq. 1 is a simplified approach/model which holds true for the many cases. (In reality MR data are complex valued and e.g. in low signal region noise is not “Rician distributed” same is true for some artifact effects which might not “so easily be modelled”. You mention this, but a small addition like “a simplified model to describe MRI bias…” would be more correct in my opinion.)

**Justification Of Final Rating:**

Overall, after re-reviewing my comments and reading other reviewers' comments I will make my initial rating final: strong accept. However, I would rather suggest a poster than an oral presentation for the submitted work. As stated initially, the proposed manuscript is sound and well written, while not being super innovative. Therefore, it's worth presenting as poster.

**Justification Of The Preliminary Rating:**

Overall, the authors present solid work. The level of innovation is not very high, but the submitted manuscript addresses an existing and important challenge in MRI which can be (partly) tackled by deep-learning based approaches. In general, I suggest to be more critical with the own work and put it into some more context of other approaches and "gold standards".

**Questions To Address In The Rebuttal:**

Please, be a bit more critical in your discussion and outline the limits of the presented approach more clearly.

**Special Issue:**

No

---

> ### Author Response · Authors · 2024-03-12
>
> 1- Please, be a bit more critical in your discussion and outline the limits of the presented approach more clearly.
>
> We added the limitations of the proposed method in the discussion:
> “We have trained our method using only T1 and T2 MR images to demonstrate its effectiveness. We are aware that the N4 method is by default a more general method than the proposed method. In the future, we will train UBC with all sorts of MR images (FLAIR, etc.) to get a more general solution.“

---

### Official Review · Reviewer_JBYg · 2024-03-01

**Confidence:** 5
**Preliminary Rating:** 4
**Recommendation:** Poster
**Final Rating:** 4

**Summary:**

Authors propose a deep learning-based approach for bias removal. The
critical part of the proposed method is that the training is
unsupervised. Training aims to decrease the entropy of the debiased
image. When combined with regularizing losses, the minimization
yielded a powerful bias removal technique that is as accurate as N4
yet much much faster, which are demonstrated on brain MRI.

**Strengths:**

1. The method seems sound, simple and effective.
2. The problem is quite relevant. Improving speed of bias removal
   would have impact on the larger pipelines.
3. Experiments are concise but show the advantages of the method.

**Weaknesses:**

1. Authors mention "thoughtful assignment of weights", which should be
   detailed. The need for fine-tuning weights for different
   applications and images is not detailed.
2. Demonstration of the model for different sequences and applications
   is not provided. Only T1 images are shown.
3. While application of a trained model to in-domain images is shown
   to outperform N4, the general applicability of N4 is not
   acknowledged. Generalization capabilities of the proposed model are
   not demonstrated.

**Detailed Comments:**

Beyond the comments I mentioned above, I would like to highlight
another point here. Other reviewers may point out that the method is
not compared to other deep learning-based methods as a weakness. While
this can be indeed seen as a weakness, I argue that this is not a huge
weakness for this article. This model here is trained in an
unsupervised way, which makes it more comparable to N4 and other
unsupervised methods.

**Justification Of Final Rating:**

I appreciate authors' responses. I think this is a valuable contribution to the conference. I believe it would be a great contribution as a poster so that participants can discuss in length the article as well as the general question about the value of non-deep learning and deep learning methods from the perspective of general applicability and performance.

**Justification Of The Preliminary Rating:**

I believe the paper proposes an elegant method. Authors can address some of the issues in the rebuttal phase to improve the article. I provide a weak accept in the anticipation that some of the issues will be addressed by the authors in the rebuttal.

**Questions To Address In The Rebuttal:**

- Generalization capabilities of the proposed model could be
  demonstrated to strengthen the article.
- Application of the method on images from different sequences and
  anatomies would be very helpful for a better comparison.
- Limitations of the model in comparison to N4 could be better
  discussed.

**Special Issue:**

No

---

> ### Author Response · Authors · 2024-03-12
>
> 1- Authors mention "thoughtful assignment of weights", which should be detailed. The need for fine-tuning weights for different applications and images is not detailed.
>
> This part was very important since image entropy solely cannot be used because this may lead to a null image (i.e. 0 entropy). Therefore, the statistical constraints (i.e. lambda4 is the most important constraint followed by lambda3 and lambda2). The optimal values were experimentally found, being lambda1 and lambda4 the most important ones.
>
> 2- Demonstration of the model for different sequences and applications is not provided. Only T1 images are shown.
>
> Following the reviewer request we have added results for T2 images. As expected, due the unsupervised nature of the proposed method, the results were coherent with those obtained for the T1 images.
>
> 3- While application of a trained model to in-domain images is shown to outperform N4, the general applicability of N4 is not acknowledged. Generalization capabilities of the proposed model are not demonstrated.
>
> We have modified the paper to acknowledge the general applicability of N4 method and the limitations of the proposed approach.
>
> “We have trained our method using only T1 and  T2 MR images to demonstrate its effectiveness. We are aware that the N4 method is by default a more general method than the proposed method. In the future, we will train UBC with all sorts of MR images (FLAIR, etc.) to get a more general solution. “
>
> 4- Generalization capabilities of the proposed model could be demonstrated to strengthen the article.
>
> To demonstrate the generalization capabilities of the proposed method, we have also trained with T2 images. The results demonstrate that the proposed method can deal with other images modalities. In the future, we plan to train our proposed method with all sorts of MR image sequences.
>
> 5- Application of the method on images from different sequences and anatomies would be very helpful for a better comparison.
>
> We have trained and tested the proposed method also with T2 images showing its competitive performance. In the future, we plan to train our proposed method with all sorts of MR image sequences.
>
> 6- Limitations of the model in comparison to N4 could be better discussed.
>
> We have modified the paper to acknowledge the general applicability of N4 method and the limitation of the proposed approach.
>
> “We have trained our method using only T1 and T2 MR images to demonstrate its effectiveness. We are aware that the N4 method is by default a more general method than the proposed method. In the future, we will train UBC with all sorts of MR images (FLAIR, etc.) to get a more general solution.“

---

### Meta-Review · Area_Chair_Z3H5 · 2024-04-03

**Recommendation:** Accept (Poster)
**Confidence:** 5

**Metareview:**

All reviewers agreed that the problem of automatic MR image bias correction using the proposed method is interesting. While the experimental results seemed promising, one reviewer suggested that conducting a comprehensive comparison with existing methods would enhance the robustness and validity of the proposed approach. Given the consistent positive recommendations from all reviewers, the meta reviewer is pleased to recommend acceptance of the paper. However, the authors are strongly encouraged to thoroughly address all comments raised by the reviewers in their final revisions.

---

### Decision · Program_Chairs · 2024-04-05

Accept (Poster)